# Novel Chemical Architectures Based on Beta-Cyclodextrin Derivatives Covalently Attached on Polymer Spheres

**DOI:** 10.3390/polym13142338

**Published:** 2021-07-16

**Authors:** Stefan Bucur, Ionel Mangalagiu, Aurel Diacon, Alexandra Mocanu, Florica Rizea, Raluca Somoghi, Adi Ghebaur, Aurelian Cristian Boscornea, Edina Rusen

**Affiliations:** 1Faculty of Chemistry, Alexandru Ioan Cuza University of Iasi, 11 Carol 1st Bvd, 700506 Iasi, Romania; bucurm.stefan@gmail.com (S.B.); ionelm@uaic.ro (I.M.); 2Institute of Interdisciplinary Research—CERNESIM Centre, Alexandru Ioan Cuza University of Iasi, 11 Carol I, 700506 Iasi, Romania; 3Faculty of Applied Chemistry and Materials Science, University Politehnica of Bucharest, 1-7 Gh. Polizu Street, 011061 Bucharest, Romania; aurel_diacon@yahoo.com (A.D.); mocanu_alexandra85@yahoo.com (A.M.); flori_rizea@yahoo.com (F.R.); adi.ghebaur@yahoo.com (A.G.); cristian.boscornea@yahoo.com (A.C.B.); 4National Research and Development Institute for Chemistry and Petrochemistry—ICECHIM, 202 Splaiul Independenţei, 060021 Bucharest, Romania; raluca.somoghi@yahoo.com; 5Advanced Polymer Materials Group, University Politehnica of Bucharest, Gh. Polizu Street, 011061 Bucharest, Romania

**Keywords:** beta-cyclodextrin, emulsion polymerization, adsorption kinetics, adsorption isotherms, bisphenol A adsorption

## Abstract

This study presents the synthesis and characterization of polymer derivatives of beta-cyclodextrin (BCD), obtained by chemical grafting onto spherical polymer particles (200 nm) presenting oxirane functional groups at their surface. The polymer spheres were synthesized by emulsion polymerization of styrene (ST) and hydroxyethyl methacrylate (HEMA), followed by the grafting on the surface of glycidyl methacrylate (GMA) by seeded emulsion polymerization. The BCD-polymer derivatives were obtained using two BCD derivatives with hydroxylic (BCD-OH) and amino groups (BCD-NH_2_). The degree of polymer covalent functionalization using the BCD-OH and BCD-NH_2_ derivatives were determined to be 4.27 and 19.19 weight %, respectively. The adsorption properties of the materials were evaluated using bisphenol A as a target molecule. The best fit for the adsorption kinetics was Lagergren’s model (both for Q_e_ value and for R^2^) together with Weber’s intraparticle diffusion model in the case of ST-HEMA-GMA-BCD-NH_2_. The isothermal adsorption evaluation indicated that both systems follow a Langmuir type behavior and afforded a Q_max_ value of 148.37 mg g^−1^ and 37.09 mg g^−1^ for ST-HEMA-GMA-BCD-NH_2_ and ST-HEMA-GMA-BCD-OH, respectively. The BCD-modified polymers display a degradation temperature of over 400 °C which can be attributed to the existence of hydrogen bonds and BCD thermal degradation pathway in the presence of the polymers.

## 1. Introduction

Bisphenol A (BPA) is one of the most abundant chemical synthetic additives produced for the manufacturing of epoxy resins, polysulfones, unsaturated polyesters, polyacrylate resins polycarbonate plastics, and rubber [1,2]. Human exposure to BPA [3,4] or its bisphenol-based substitutes [5] exhibited cytotoxicity, neurotoxicity, endocrine-disrupting effects, reproductive toxicity, uterine cancer, and interference of cellular pathways, capable of mimicking some of the hormones of the human body [1,3,4,5]. A recent study [6] revealed that the estimated intake of BPA is 30.76 ng/kg per body weight per day, as this compound enters in various environmental media (air, soil, aquatic systems) and the food chain due to improper recycling procedures [1,7]. This value, although in some countries did not exceed the maximum acceptable values for daily intake, is all the more worrying as the first six countries with the highest values belong to the European Union [6]. Thus, it is of tremendous importance on the one hand to develop efficient decontamination procedures and on the other hand to impose stricter laws regarding plastic manufacturing or recycling. 

Different methods for BPA removal were directed to biological treatment, membrane adsorption processes, or advanced oxidation [1]. The biological treatment involves the degradation of BPA by immobilization of enzymes such as oxidoreductase or polyphenol oxidases [8] that are capable of oxidizing organic pollutants, such as phenols, organic dyes, or drugs. Advanced oxidation is a method that removes BPA by the generation of highly reactive radicals or the application of photocatalytic treatment to break down the molecules of BPA into less harmful compounds from water, sediment, and soil [1,9,10,11]. The membrane adsorption technology is based on the selective adsorption capacity of different membranes of targeted pollutants (by chemical or physical interaction) and is sometimes preferred for BPA removal since it does not generate new harmful by-products [12,13]. 

The drawbacks of these methods are not only related to the potential of generating harmful by-products but also to the difficulty of controlling the necessary parameters for efficient removal of BPA (such as pH and temperature in the case of enzymes or photocatalytic removal) and laborious/expensive synthesis methods for adsorption membranes [2]. 

Cyclodextrins (CDs) are a class of three-dimensional (3D) cyclic oligosaccharides composed of d-glucose units linked together by α-1,4-glucosidic bonds with hydrophilic surface and hydrophobic internal hollow [14,15]. The amphiphilic, biodegradable, and non-hazardous properties of such compounds, as well as the possibility to design new cyclodextrin derivatives, has led to a wide range of applications for drug delivery systems, antiviral therapy, cosmetics, agriculture, enzymology, catalysis, enantiomers separation, and environmental protection, in which it has been shown they can be excellent candidates for decontamination of aqueous media, air or soil [15,16,17,18,19]. 

As environmental issues are becoming more important, intense research interest has been dedicated to the removal of pollutants from water using different cyclodextrins-polymer systems that are tailored to eliminate organic or inorganic pollutants based on the adsorption process of the targeted molecule [20,21,22]. Based on the 3D spatial structure of CDs and on the hydrophilic/hydrophobic building, the enhanced removal of pollutants is based on guest-host interactions between the cyclodextrin derivative and the targeted pollutant. The additional presence of polymer chains can improve the adsorption capacity of the contaminating agent depending on their chemical structure or morphology, since polymer adsorbents can include polymers with a spherical shape, intrinsic porosity, metal/covalent-organic frameworks, as well as hyper-crosslinked polymers [23,24,25]. 

Thus, literature data is abundant in different synthesis approaches that have the same goal, to improve the adsorption efficiency of the cyclodextrins-polymers (CDP) systems for wastewater decontamination. For example, beta-cyclodextrin (BCD) was reacted with epichlorohydrin (EPI) and further with trimesoyl chloride to obtain a CDP system that was further embedded by interfacial cross-linking into a nylon microfiltration membrane to create a porous structure with enhanced adsorption capacities of water contaminants [26]. The adsorption efficiency capacity of modified BCD is amazing considering that structurally different contaminants like BPA, methylene blue, and copper can be adsorbed simultaneously by using citric acid-crosslinked-BCD polymers [27]. Recent approaches demonstrated that BCD-based polymers obtained by crosslinking with EPI can be used in municipal wastewater treatment pilot plants to remove several micropollutants, including BPA, with over 80% efficiency [28]. Also, the BCDP adsorption capacities were improved by the presence of nano-adsorbents such as Fe_3_O_4_, SiO_2_, silver nanoparticles, or carbon nanotubes due to their high specific surface area and absence of internal diffusion resistance that enhances the kinetics for the adsorption processes of different contaminating agents such as BPA, organophoshorous insecticides, or p-nitrophenol from water [9,20,29]. 

Thus, in this work, novel BCD modified polymer particles with spherical morphology were synthesized for a possible decontamination process of wastewater. Utilizing the oxirane functional groups present at the surface of polymer particles, two types of BCD derivatives (with hydroxylic (BCD-OH) and amino groups (BCD-NH_2_)—Scheme 1) were chemically grafted to the polymer colloids (Scheme 2). The materials designed aimed to improve the interaction between the polymeric adsorbent, pollutant, and the contaminated media. One of the main goals of this study was to determine the linking capacities of the BCD to the polymer, as well as the maximum complexation capacity of the BCD-modified polymers toward a targeted molecule. Thus, BPA was selected as a model molecule to investigate the adsorption kinetics, complexation mechanism, and isotherms. 

## 2. Materials and Methods

### 2.1. Materials

Styrene (ST) (Sigma-Aldrich, St. Louis, MO, USA) has been purified through vacuum distillation. 2-Hydroxyethyl methacrylate (HEMA) (Sigma-Aldrich, St. Louis, MO, USA) and glycidyl methacrylate (GMA) (Merck, Darmstadt, Germany) have been purified by passing over short columns of activated basic alumina. Potassium persulfate (K_2_S_2_O_8_) (KPS) (Merck, Darmstadt, Germany) has been recrystallized from an ethanol/water mixture and then vacuum-dried. β-cyclodextrin (BCD-OH) (≥95.0%, Wacker Chemie, Munich, Germany) was vacuum-dried before use for 24 h. p-Toluene sulfonyl chloride (Ts-Cl) (reagent grade, ≥98%, Merck, Darmstadt, Germany), sodium hydroxide (reagent grade, ≥98%, pellets (anhydrous), (Sigma-Aldrich, St. Louis, MO, USA), 1,4-Diaminobutane (99%, Sigma-Aldrich, St. Louis, MO, USA), acetone (for analysis, >99%, Chemical Company Iasi), dimethyl sulfoxide (DMSO) (Aldrich-anhydrous, Darmstadt, Germany), pyridine (Aldrich, Darmstadt, Germany), glucose (Aldrich anhydrous, Darmstadt, Germany), phenol (Merck, Darmstadt, Germany), sulfuric acid (Sigma-Aldrich, St. Louis, MO, USA), bisphenol A (Merck, Darmstadt, Germany), ethanol (Chimopar, Bucuresti, România) were used as received.

### 2.2. Methods

#### 2.2.1. ST-HEMA Emulsion Polymerization

A mixture of ST (1.3 mL), respectively, HEMA (0.3 mL) and KPS (25 mg) was added to distilled water (20 mL). The mixture was purged with nitrogen and then maintained for 8 h at 75 °C under continuous stirring. The final dispersion was dialyzed in distilled water for 7 days, using cellulose dialysis membranes (molecular weight cut-off: 12,000–14,000), to remove the unreacted monomers and initiator. 

#### 2.2.2. Seeded Emulsion Polymerization of GMA (ST-HEMA-GMA)

To the dialyzed emulsion presented previously was added 0.3 mL GMA and KPS (25 mg). The mixture was purged with nitrogen and then maintained for 8 h at 75 °C under continuous stirring. The final dispersion was dialyzed in distilled water for 7 days, using cellulose dialysis membranes (molecular weight cut-off: 12,000–14,000), to remove the unreacted monomers and initiator.

#### 2.2.3. Synthesis of Diamino Butane Monosubstituted BCD (BCD-NH_2_)

The Ts-BCD was synthesized by a method similar to that reported by Brady et al. [30] (For the NMR spectra and Maldi see Appendix A). The procedure for obtaining BCD-NH_2_ was as follows: 5.5 g Ts-BCD (4.266 mmol) were dissolved in 166 mL of 1,4-diaminobutane (DAB), slowly warmed up to 70 °C, and kept at this temperature for 24 h. At the end of reaction time, the solvent was vacuum distilled and the solid resulted was dissolved in a minimum amount of water. This syrup was added dropwise into 150–200 mL of acetone and precipitates a white solid. At least two acetone precipitations are needed to obtain a white powder solid, otherwise the product seems oily. The final product was obtained by drying the sample in a vacuum oven at 40 °C for 2 days with a 54% yield. **^1^H-NMR** (DMSO-d6, 500 MHz): δ 1.405 (m, H-8, H-9), 2.5–2.562 (s, NH_2_, NH, DMSO), 2.688 (m, H-7), 2.882–2.860 (m, H-10), 3.346–3.304 (m, H-2, H-4), 3.627–3.551 (m, H-3, H-5, H-6), 4.474 (s, 6-OH), 4.823 (s, H-1), 5.736 (br, 2-OH and 3-OH) and **^13^C NMR** (DMSO-d6, 125 MHz): δ 26.94 (C-9), 29.33 (C-8), 40.69 (C-10), 49.05 (C-7), 49.40 (C-6), 59.93 (C-6*), 72.05 (C-5), 72.43 (C-2), 73.07 (C-3), 81.55 (C-4), 101.96 (C-1). (see Appendix A)).

**MALDI:** calc. for C_46_H_80_N_2_O_34_, M is 1204.45 Da (monoisotopic mass); the simple [M]^+^ or [M+H]^+^ were not identified but the mass 1215 corresponds to a complex adduct [2M+Na]^2+^.

#### 2.2.4. The Reaction of BCD-OH with ST-HEMA-GMA

The ST-HEMA-GMA emulsion was dried in an oven at 70 °C on a glass plate. A solid powder was obtained after the water evaporation. 0.5 g ST-HEMA-GMA powder was dispersed in 10 mL DMSO and heated to 80 °C. After 30 min, 0.6 g BCD-OH was added to the mixture together with 0.01 mL pyridine, as catalysis. The reaction was kept to the temperature for 12 h and precipitated into hot water. The obtained white powder was filtered and dried in the oven.

#### 2.2.5. The Reaction of BCD-NH_2_ with ST-HEMA-GMA

0.5 g ST-HEMA-GMA powder was dispersed in 10 mL DMSO and heated to 80 °C. After 30 min, 0.7 g BCD-NH_2_ was added to the mixture together with 0.01 mL pyridine, as catalysis. The reaction was kept to the temperature for 12 h and precipitated into hot water. The obtained white powder was filtered and dried in the oven.

### 2.3. Characterization

The morphology of the polymer nanoparticles was evaluated using transmission electron microscopy (TEM) (Tecnai G2 F20 TWIN Cryo-TEM, FEI Company), at 300 kV acceleration voltage, at a 1 Å resolution. 

FTIR spectra were recorded on a Bruker VERTEX 70 spectrometer using 32 scans with a resolution of 4 cm^−1^ in 4000–600 cm^−1^ region. The samples were analyzed using the attenuated total reflection (ATR) technique.

The UV-Vis spectra were recorded using a V-550 Able Jasco spectrophotometer, using a bandwidth of 1 nm, and a scanning speed of 1000 nm min^−1^. The BCD content was quantified by determining the reducing sugars of the polymer using concentrated H_2_SO_4_ acidolysis and phenol colorimetric analysis [31]. The procedure employed for the determination of BCD content and the calibration curve are presented in the Appendix A.

The thermogravimetric analyses (TGA) were performed using a Netzsch TG 209 F3 Tarsus equipment considering the next parameters: nitrogen atmosphere flow rate 20 mL min^−1^; samples mass: ∼3 mg; temperature range: room temperature −700 °C; heating rate: 10 °C min^−1^ in an alumina crucible.

NMR experiments were carried out on Bruker Avance III 500, ^1^H NMR spectra were recorded at 500 MHz using the solvent peaks as internal references and ^13^C NMR spectra were recorded at 125 MHz. All NMR experiments were conducted according to the literature, and they can be found in the Appendix A.

Matrix-assisted laser desorption ionization time of flight (MALDI-TOF) mass spectrometry experiments were carried out on Shimadzu AXIMA Performance, operated in high-resolution reflectron mode using α-Cyano-4-hydroxycinnamic acid as matrix.

The fluorescence spectra have been registered using a FP-6500 Able Jasco spectrofluorometer.

## 3. Results and Discussion

The first stage of our study consisted in the synthesis of polymeric particles presenting the capacity for the chemical attachment of BCD derivatives (BCD-OH and BCD-NH_2_, Scheme 1). The spherical shape was selected due to its high specific surface and high surface-to-volume ratio. The presence of an oxirane functional group at the surface of the polymer particles also permits the grafting of both beta-cyclodextrin derivatives (BCD-OH and BCD-NH_2_). Therefore, the synthesis strategy involved the emulsion polymerization of ST-HEMA system [32], followed by the seeded polymerization of GMA on the surface of the polymer particles [33]. As a result, an oxirane functional group capable of reacting with both hydroxyl and amino functional groups will be present on the surface of the polymer particles after the seeded polymerization. Figure 1 presents TEM images of the polymer particles at different stages during the synthesis route: ST-HEMA (after the emulsion polymerization), ST-HEMA-GMA (after the seeded polymerization), and ST-HEMA-GMA-BCD-NH_2_ (after BCD-NH_2_ grafting to the polymeric particle).

From the analysis of Figure 1, it can be observed that the ST-HEMA polymer particles are around 200 nm with a monodisperse size distribution. The average particle size and the size distribution are increased after the GMA seeded polymerization. Thus, the average particle size increases to around 220–230 nm, and a core-shell structuring can be noticed (inset detail Figure 1b). In the case of BCD-NH_2_ and BCD-OH grafting on the surface of the polymer particles, aggregates with an average dimension of 20 nm [34] can be observed deposited on the polymer particles after the seeded polymerization (darker spheres around the ST-HEMA-GMA Figure 1c,d).

To sustain the chemical grafting of the BCD derivative through the oxirane group reaction, the materials were analyzed by FT-IR spectroscopy (Figure 2). Thus, the specific vibration of oxirane 907 cm^−1^ decreased while the intensity of the signal at 3420 cm^−1^_,_ specific for -OH vibration increased. Moreover, the appearance of C-O-C vibration signal at 1033 cm^−1^ and 3266 cm^−1^ signal specific for NH vibration confirm the chemical attachment of the BCD derivatives to the polymeric particles. The signal at 1026 cm^−1^ specific for the vibration of primary alcohol functional groups can also be identified in the ST-HEMA-GMA-BCD-OH and ST-HEMA-GMA-BCD-NH_2_ samples. In addition, the C−N bond that is formed during BCD-NH_2_ grafting to the polymer particles can be highlighted in the BCD-NH_2_ and ST-HEMA-GMA-BCD-NH_2_ samples at 1143 cm^−1^.

After the BCD attachment to the surface of the polymer particles, the degree of grafting was determined by NMR, TGA, and UV-Vis spectroscopy (see Appendix A—calibration curve and determination method). Thus, BCD-OH content was determined at 4.27%, while in the case of BCD-NH_2_ the content was significantly higher at 19.19%. The difference in BCD content in the final materials can be explained by the higher reactivity of the oxirane group towards the amino than hydroxyl groups. 

The organic pollutants present in water, such as bisphenol A, can be removed relatively easily and at a low cost by adsorption processes that can exploit the presence of BCD cage structure immobilized on polymeric supports [35]. In this study, we have determined the quantity of bisphenol A adsorbed per gram of material using ST-HEMA-GMA-BCD materials (Q_e_ (mg bisphenol A/g polymer)). Thus, using a calibration curve for fluorescence intensity depending on the bisphenol A concentration (Appendix A), the amount adsorbed onto the BCD modified polymer spheres was determined using Equation (1): (1)$$Q_{e}=\frac{\left(c_{0}-c_{e}\right)\times V}{m}$$
where V is the solution volume (mL), c_0_ (mg L^−1^) and c_e_ (mg L^−1^) are the initial and final solution concentrations of bisphenol A and m is the mass of BCD modified polymer particles (mg).

After 240 min of interaction at 25 °C, the adsorption capacity Q_e_ values obtained were 9.51 and 12.57 mg/g for ST-HEMA-GMA-BCD-OH and ST-HEMA-GMA-BCD-NH_2_, respectively. The higher Q_e_ value obtained for the polymers modified using the BCD-NH_2_ can be related to the higher BCD grafting degree. It is easily observed that the large difference in grafting efficiency is in contrast with the relatively small difference between the adsorption capacity at equilibrium. This can be explained by the formation of hydrogen bonds between the hydroxyl groups of BCD-OH and bisphenol A [20]. Nevertheless, the higher bisphenol A adsorption characteristics of ST-HEMA-GMA-BCD-NH_2_ are evident. Consequently, a kinetic analysis of the adsorption process was performed for ST-HEMA-GMA-BCD-NH_2_ (Figure 3).

Figure 3 illustrates the adsorption of bisphenol A onto the ST-HEMA-GMA-BCD-NH_2_ as a function of time in contact. It can be noted that the absorption rate is fast up to 60 min which is followed by a slight decrease up to 200 min, followed by an equilibrium tendency of the system in the 200–300 min. The first stage can be explained by a high bisphenol A concentration in solution which diffuses to the polymer particles surface. The second stage, from 60 to 200 min, represents intermediary behavior during which the adsorption rate decreases as the internal diffusion resistance increases, which is finally followed by the equilibrium characteristics after 200 min. Several mathematical models were analyzed to determine the adsorption efficiency and the mechanism that it follows: Lagergren’s pseudo-first-order kinetic model (Equation (2)), Ho’s pseudo-second-order model (Equation (3)), and Weber’s intra diffusion model (Equation (4)):(2)$$\ln\left(Q_{e}-Q_{t}\right)={\mathrm{lnQ}}_{e}-k_{1}t$$
(3)$$\frac{t}{Q_{t}}=\frac{1}{k_{2}Q_{e}^{2}}+\frac{t}{Q_{e}}$$
(4)$$Q_{t}=K_{p}\sqrt{t}$$
where Q_e_ (mg g^−1^) and Q_t_ (mg g^−1^) are the amounts of bisphenol A adsorbed per unit mass of ST-HEMA-GMA-BCD-NH_2_ at equilibrium and t (min), respectively. k_1_ is the pseudo-first-order adsorption rate constant (min^−1^), k_2_ is the pseudo-second-order adsorption rate constant (g (mg min)^−1^), and K_p_ is the intraparticle diffusion constant (mg g^−1^ min^−0.5^).

The kinetic parameters of Lagergren’s pseudo-first-order, Ho’s pseudo-second-order, and Weber’s intra particles diffusion models equations were calculated by slope-intercept of the linear fitting plots of ln(Q_e_-Q_t_) versus t, t/Q_t_ versus t, and Q_t_ versus t, respectively (Figure 4).

Comparing the experimental data with the results from the mathematical models for the adsorption process, the best fit was Lagergren’s (both for Q_e_ value and for R^2^) together with Weber’s intraparticles diffusion model for the step that controls the mass transfer (Figure 3 and Table 1). Analyzing the values for K_p_ of Weber’s model, this corresponds to a rapid adsorption process in the first stage of contacting [36].

The study of the adsorption isotherms offers information on the interaction between the adsorbate and the adsorbent and allows the determination of the adsorption capacity of the adsorbent, which is an important parameter for system evaluation. The most intensively used isotherm adsorption model are Langmuir (Equation (5)) and Freundlich (Equation (6)).
(5)$$\frac{1}{Q_{e}}=\frac{1}{Q_{\max}}+\frac{1}{Q_{m}K_{L}}\times \frac{1}{c_{e}}$$
(6)$${\mathrm{lnQ}}_{e}={\mathrm{lnK}}_{F}+\frac{1}{n}{\mathrm{lnc}}_{e}$$
where Q_e_ (mg g^−1^) indicates the amount of adsorbate at equilibrium; Q_max_ (mg g^−1^) indicates the maximum amount of adsorbate at equilibrium; c_e_ (mg L^−1^) is the equilibrium concentration of the adsorbate in the solution; K_L_ (L mg^−1^) and K_F_ (L mg^−1^) are the Langmuir and Freundlich constants, respectively; and n is the heterogeneity factor.

The isothermal adsorption at the equilibrium of bisphenol A by ST-HEMA-GMA-BCD-NH_2_ and ST-HEMA-GMA-BCD-OH are represented in (Appendix A). From this, it can be noted that the adsorption capacity increases with the increase of adsorbate and reaches saturation as the bisphenol A concentration exceeds 45 mg/L and 40 mg/L in the case of ST-HEMA-GMA-BCD-NH_2_ and ST-HEMA-GMA-BCD-OH, respectively. The linearization of the two selected isotherms Equation (5), Equation (6), and the parameters calculated are presented in Figure 5 and Table 2.

The comparison of the Langmuir and Freundlich isotherm parameters as listed in Table 2 indicates better linearity in the case of the Langmuir plots, suggesting that the adsorption of bisphenol A by ST-HEMA-GMA-BCD-NH_2_ and ST-HEMA-GMA-BCD-OH can be regarded as a monolayer adsorption process. This means that the adsorption sites located on the surface of the polymer particles were homogeneously distributed and an equivalent adsorption force is displayed. Additionally, it could be evaluated from the Langmuir equation that the Q_max_ for bisphenol A was 148.37 mg g^−1^ and 37.09 mg g^−1^ for ST-HEMA-GMA-BCD-NH_2_ and ST-HEMA-GMA-BCD-OH, respectively. (Table 3) For a cross-linked polymer, three main adsorption mechanisms were proposed which involve (1) host–guest interactions in the polymers cavities, (2) interactions in the pores of the polymeric network, and (3) interactions on the surface (physical sorption) [37]. The difference between the adsorption characteristics of the two polymers (Q_max_) can be correlated to the amount of BCD grafted to the polymer particles [38]. Thus, due to its higher reactivity, BCD-NH_2_ afforded a larger substitution degree which is reflected by the enhanced Q_max_ value.

Comparing the values for Q_max_ for the adsorption of bisphenol A using different adsorbents (Table 3), it can be noted that the values obtained for our materials are reasonably high. The increased efficiency is due to the high ratio between the specific surface to the mass of the polymer particles and the high reactivity of the surface functional groups (oxirane). These characteristics permit the synthesis of BCD functionalized polymer particles with a good weight ratio (BCD to polymer support) and a proper distribution of BCD on the surface of the polymer particles. Therefore, the possibility of guest–host interaction is facilitated from the steric point of view (see Appendix A). In addition to this interaction, there is also the possibility for adsorption processes in the pores of the cross-linked polymer particles, respectively physical interaction on the surface of the materials [20]. To confirm the interaction between the polymeric structures modified with BCD and bisphenol A (BPA) FTIR analysis was performed on the materials after the adsorption evaluation. From the spectra (see Appendix A Appendix A), the appearance of a novel signal at 1150 cm^−1^ corresponding to the C-O vibration from BPA [45] can be observed, confirming the presence of BPA on the surface of the polymer particles.

The thermal resistance is one of the most important properties of polymers and their derivatives, in our case the polymer particles modified with BCD-OH and BCD-NH_2_. Figure 6 presents the TGA and DTG curves for the synthesized polymers and starting materials. 

Comparing the temperatures corresponding for a 10% weight loss (Table 4) it can be noted that BCD-OH has the highest thermal resistance, followed by ST-HEMA-GMA-BCD-OH and ST-HEMA-BCD-NH_2,_ possibly due to the hydrogen bonds formed between the -OH groups. The ST-HEMA and ST-HEMA-GMA display lower thermal stability. The TGA analysis also establishes the temperature where the maximum weight loss takes place. In this case, the T_max_ indicated that the most stable materials were ST-HEMA-GMA-BCD-OH and ST-HEMA-GMA-BCD-NH_2,_ which display stability up to 400 °C.

## 4. Conclusions

A monoamine derivative of BCD was synthesized, characterized by NMR and mass spectroscopy. The BCD derivative was chemically grafted to ST-HEMA-GMA particles obtained by seeded emulsion polymerization. The GMA was used during the seeded polymerization due to its high reactivity towards -OH and NH_2_ functional groups. The chemical attachment of the BCD derivatives (BCD-OH and BCD-NH_2_) to the surface of the polymer particles was qualitatively confirmed by FT-IR spectroscopy and quantitatively by acidolysis and phenol colorimetric analysis. The degree of polymer functionalization using the BCD-OH and BCD-NH_2_ derivatives were 4.27 and 19.19 weight %. 

The adsorption properties of the materials were evaluated using bisphenol A as target molecule. The results from the mathematical models for the adsorption kinetics process indicated that the best fit was Lagergren’s model (both for Q_e_ value and for R^2^), together with Weber’s intraparticles diffusion model for the step that controls the mass transfer in the case of ST-HEMA-GMA-BCD-NH_2_. The isothermal adsorption evaluation indicated that both systems follow a Langmuir type behavior and afforded a Q_max_ value of 148.37 mg g^−1^ and 37.09 mg g^−1^ for ST-HEMA-GMA-BCD-NH_2_ and ST-HEMA-GMA-BCD-OH, respectively. Due to the higher reactivity of BCD-NH_2,_ a larger substitution degree of the polymer particles was obtained, which is reflected by the enhanced Q_max_ value.

The thermogravimetric analysis of the materials indicated that the functionalization of the polymer particles afforded an increase of the thermal resistance. Thus, the BCD modified polymers present a degradation temperature of over 400 °C, which can be attributed to the hydrogen bonds and BCD thermal degradation pathway in the presence of the polymers.

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
