# Peer review of "Novel Chemical Architectures Based on Beta-Cyclodextrin Derivatives Covalently Attached on Polymer Spheres"

_polymers, 2021, doi:10.3390/polym13142338_

Round 1

Reviewer 1 Report

In this manuscript by Bucur et al., the authors present results from their study evaluating the adsorption of bisphenol A (BPA) from waste water to polymer derivatives of beta-cyclodextrin (BCD), chemically grafted onto spherical polymer particles. In particular the authors synthesized adsorbents with two BCD derivatives with a hydroxyl and amino groups in their lab. Reasonably good performance was observed for both adsorbents and both adsorbents showed thermal stability up to 400 degree C. The study is original and carried out carefully. In general the manuscript is well-written too. I therefore recommend accepting this manuscript for publication after the authors have addressed the following minor issues

  1. In Figure 2, the TEM image of the sample ST-HEMA-GMA-BCD-NH2 is shown along with those of the ST-HEMA and ST-HEMA-GMA samples. However a TEM image of ST-HEMA-GMA-BCD-OH is conspicuously absent. The authors should include the images of samples with both BCD derivatives. Further, the scale in the three images 2(a), (b) and (c) seem to be different. For a fair comparison, all images should use the same scale. This is doable as the characteristic size of the particles is of the same order of magnitude.
  2. Lines 258 and 261, I suppose the authors are referring to Figure 3 and not Figure 2 as stated. This should be corrected.
  3. Figure 6 looks too crowded because of the overlapping of the weight and derivative weight curves. Perhaps the authors could shift the scale of the derivative weight to higher regions to avoid the overlap.
  4. Typo on line 139, the word 'reported' is repeated.
  5. Figure 4, the X-axis label should be √(time) (square root of time).

Author Response

Response to reviewer 1 comments

We would like to begin this reply by thanking the reviewer for taking the time to assess the manuscript. Bellow, marked in green are the responses to each observation/suggestion made by the reviewer.

Reviewer 1:

  1. In Figure 2, the TEM image of the sample ST-HEMA-GMA-BCD-NH2 is shown along with those of the ST-HEMA and ST-HEMA-GMA samples. However a TEM image of ST-HEMA-GMA-BCD-OH is conspicuously absent. The authors should include the images of samples with both BCD derivatives. Further, the scale in the three images 2(a), (b) and (c) seem to be different. For a fair comparison, all images should use the same scale. This is doable as the characteristic size of the particles is of the same order of magnitude.

  1. TEM images for ST-HEMA-GMA-BCD-OH were added in Figure 1d.
  2. Lines 258 and 261, I suppose the authors are referring to Figure 3 and not Figure 2 as stated. This should be corrected.

  1. The manuscript was corrected in accordance with reviewer’s observation.

  1. Figure 6 looks too crowded because of the overlapping of the weight and derivative weight curves. Perhaps the authors could shift the scale of the derivative weight to higher regions to avoid the overlap.

  1. Figure 6 was split into two separate images to improve the readability in accordance with the reviewer’s observation.

  1. Typo on line 139, the word 'reported' is repeated.

  1. The manuscript text was corrected.
  2. Figure 4, the X-axis label should be √(time) (square root of time).
  3. The label was corrected.

Reviewer 2 Report

The manuscript (Polymers-1284898) reported the synthesis and characterization of polymer derivatives of beta-cy-clodextrin (BCD), obtained by chemical grafting onto spherical polymer particles (200 nm) present-ing oxirane functional groups at their surface. The materials have higher adsorption capacity towards bisphenol A than the previous reports. Therefore, I think that it can meet journal standards for novelty. It is clearly stated there that studies is enough interest to the readers. Therefore, Therefore, I recommend that this manuscript should be published in Polymers after major revision.

  1. English language style and accuracy should be improved.
  2. some evidence should be goven to confirm the mechanism of interaction adsorption such as FTIR and XPS results.
  3. Some figuress and tables should be more standardized。

Author Response

Response to reviewer 2 comments

We would like to begin this reply by thanking the reviewer for taking the time to assess the manuscript. Bellow, marked in green are the responses to each observation/suggestion made by the reviewer.

Reviewer 2

  1. English language style and accuracy should be improved.
  1. The manuscript was rechecked and several modifications were made to this aspect.
  1. some evidence should be goven to confirm the mechanism of interaction adsorption such as FTIR and XPS results.
  1. FTIR analysis was performed on the polymers modified with BCD (see Figure S11 – supplementary info).
  1. Some figuress and tables should be more standardized。
  1. We have tried to keep the format of the images as consisted as possible.

Reviewer 3 Report

General impression.

The authors presented an interesting material capable of adsorption of bisphenol-A. The structure of the articles is clear and easily leads the reader through all of the data and information.

Some issues need to be fixed before the publication. The list of vital questions is given in Section 1, whereas some minor mistakes are listed in Section 2 (below).

Section 1.

I recommend correcting the following points, answer the questions, solve the problems by introducing changes in the manuscript and/or supporting information.

General

  • Please, explain the symbols of the graphs in the main text and in the supplementary. Full description of axis or naming the symbols (e.g., Qe) below the figure will fasten data extraction by the reader and ease the understanding.
  • Scheme 2. The functional groups are merely visible. Please rearrange the picture or add a zoom-in view of the groups or symbolic groups with a description.

Determination of cyclodextrin content in the polymer

  • The degree of grafting was determined by TGA, NMR, and UV-Vis. However, only the results of UV-Vis are calculated and presented with specific values. What are the other results in the context of the degree of grafting?
  • For the calibration, the glucose solution was incubated at 50 °C and then cooled down to room temperature for the measurements. Why such a temperature? Should not it be done at the same temperature as the sample? Was the sample measured also at room temperature?
  • Please explain better the origin of the formula for BCD content. What are the 180 and 0.01 standing for? (Supplementary)
  • What is the origin of the red curves in Figure S9? The presented points suggest a linear model that is tested in the next paragraph of the manuscript (with two models). In both cases, Langmuir and Freundlich, a relatively high percentage of explanation of the phenomena with those models (R2≈ 95%) was gained. According to that, why is saturation reached when ce exceed 45 and 40 mg/L while at those points Qe is much lower than calculated Qmax?
  • Both Freundlich and Langmuir isotherms models were almost equally adequate to explain the experimental results. Due to that, the results of the experiments can also suggest that the active sites of the material are one by one occupied due to the adsorption what could indicate the host-guest mechanism or interactions in the pores of the polymeric network. For example, deviation from the isotherms can be described by the physical sorption regarding the three mentioned mechanisms. According to that, can it be called Langmuir monolayer adsorption?

Section 2.

  • There is no consistency with writing the values and the units. Please separate the values and the units with space. Please read the manuscript and make it consistent.
  • Line 84. Please, explain the “CDP” abbreviation at the first use.
  • Line 116. vacuum-dried
  • Line 138. … similar to reported by Brady…
  • Line 159. the unit is missing

Author Response

Response to reviewer 3 comments

We would like to begin this reply by thanking the reviewer for taking the time to assess the manuscript. Bellow, marked in green are the responses to each observation/suggestion made by the reviewer.

Reviewer 3

  • Please, explain the symbols of the graphs in the main text and in the supplementary. Full description of axis or naming the symbols (e.g., Qe) below the figure will fasten data extraction by the reader and ease the understanding.

When first used the notation were explained. It would be redundant to repeat their designation each time.

  • Scheme 2. The functional groups are merely visible. Please rearrange the picture or add a zoom-in view of the groups or symbolic groups with a description.

Schem 2 was modified to make the functional groups more visible.

The degree of grafting was determined by TGA, NMR, and UV-Vis. However, only the results of UV-Vis are calculated and presented with specific values. What are the other results in the context of the degree of grafting?

Although both TGA and UV-Vis analysis aimed to highlight the degree of functionalization with BCD, we preferred the results from UV-Vis since in the TGA the degration of BCD derivatives partially overlaps the degradation of the polymeric backbone. Thus, UV-Vis provides a more accurate quantification of the BCD functionalization degree.

            For the calibration, the glucose solution was incubated at 50°C and then cooled down to room temperature for the measurements. Why such a temperature? Should not it be done at the same temperature as the sample? Was the sample measured also at room temperature?

In the case of the polymer derivatives a higher temperature was selected to ensure the hydrolysis and transformation of the BCD from the polymer surface. This strategy was also observed in the literature since the reactivity of the BCD one immobilized on the polymer decreases. (see 10.1039/C6RA16383A).

  • Please explain better the origin of the formula for BCD content. What are the 180 and 0.01 standing for? (Supplementary)

 Where 180 is the molecular weight of glucose, c is the glucose concentration (g/L); V is the volume of mixed solution (L); M is the molar mass of BCD (g/mol); 180 is the molecular weight of

  • What is the origin of the red curves in Figure S9? The presented points suggest a linear model that is tested in the next paragraph of the manuscript (with two models). In both cases, Langmuir and Freundlich, a relatively high percentage of explanation of the phenomena with those models (R2≈ 95%) was gained. According to that, why is saturation reached when ce exceed 45 and 40 mg/L while at those points Qis much lower than calculated Qmax?

The different values can be explained by the definition of each parameter:

Qe represents the quantity of bisphenol A adsorbed at equilibrium per gram of BCD modified polymer while the Qmax value represents the maximum amount of BPA that can be adsorbed per gram of material.

The difference between the Qmax value from Figure 5 and maximum Qe from Figure S9 could be attributed to the heterogeneity of the surface (G.D. Halsey, The role of surface heterogeneity, Adv. Catal. 4 (1952) 259–269) a characteristic of the samples that could be ascertained from the TEM images.

  • Both Freundlich and Langmuir isotherms models were almost equally adequate to explain the experimental results. Due to that, the results of the experiments can also suggest that the active sites of the material are one by one occupied due to the adsorption what could indicate the host-guest mechanism or interactions in the pores of the polymeric network. For example, deviation from the isotherms can be described by the physical sorption regarding the three mentioned mechanisms. According to that, can it be called Langmuir monolayer adsorption?

We believe that the model that best fits the materials behaviour is Langmuir monolayer adsorption that could be explained by a good distribution of the active site BCD derivatives on the surface of the polymer support.

  • Line 84. Please, explain the “CDP” abbreviation at the first use.

The CDP term explanation was added in manuscript

  • Line 116. vacuum-dried

The manuscript was modified in accordance with the reviewer’s comments.

  • Line 138. … similar to reported by Brady…
  • The manuscript was modified in accordance with the reviewer’s comments.
  • Line 159. the unit is missing
  • The manuscript was modified in accordance with the reviewer’s comments.

Round 2

Reviewer 2 Report

The current revision meets the requirements of Polymers.
